# Regression plane concept for analysing continuous cellular processes with machine learning

Abel Szkalisity[1,2], Filippo Piccinini [3], Attila Beleon[1], Tamas Balassa[1], Istvan Gergely Varga [4], Ede Migh[1], Csaba Molnar[1], Lassi Paavolainen[5], Sanna Timonen[5], Indranil Banerjee[6], Elina Ikonen [2], Yohei Yamauchi [7], Istvan Ando[4], Jaakko Peltonen[8,9], Vilja Pietiäinen [5], Viktor Honti[4] & Peter Horvath [1,5,10 ✉]

Biological processes are inherently continuous, and the chance of phenotypic discovery is significantly restricted by discretising them. Using multi-parametric active regression we introduce the Regression Plane (RP), a user-friendly discovery tool enabling class-free phenotypic supervised machine learning, to describe and explore biological data in a continuous manner. First, we compare traditional classification with regression in a simulated experimental setup. Second, we use our framework to identify genes involved in regulating triglyceride levels in human cells. Subsequently, we analyse a time-lapse dataset on mitosis to demonstrate that the proposed methodology is capable of modelling complex processes at infinite resolution. Finally, we show that hemocyte differentiation in Drosophila melanogaster has continuous characteristics.

[1] Synthetic and Systems Biology Unit, Biological Research Centre (BRC), Szeged, Hungary. [2] Department of Anatomy and Stem Cells and Metabolism Research Program, Faculty of Medicine, University of Helsinki, Helsinki, Finland. [3] Istituto Scientifico Romagnolo per lo Studio e la Cura dei Tumori (IRST) IRCCS, Meldola, FC, Italy. [4] Institute of Genetics, Biological Research Center (BRC), Szeged, Hungary. [5] Institute for Molecular Medicine Finland-FIMM, Helsinki Institute of Life Science-HiLIFE, University of Helsinki, Helsinki, Finland. [6] Indian Institute of Science Education and Research (IISER), Mohali, India. [7] School of Cellular and Molecular Medicine, University of Bristol, BS8 1TD University Walk, Bristol, UK. [8] Faculty of Information Technology and Communication Sciences, Tampere University, FI-33014 Tampere University, Tampere, Finland. [9] Department of Computer Science, Aalto University, Aalto, Finland. [10] Single-Cell Technologies Ltd., Szeged, Hungary. ✉email: horvath.peter@brc.hu

large-scale imaging technologies, such as high-content screening (HCS) and digital pathology imaging, have become the de facto tools for discovering drugs and genes and understanding tissue physiologies and pathologies, including cancer heterogeneity. This has induced a rapid growth in the amount of microscopy data, making it essential to elaborate appropriate bioinformatics tools to analyze them, and thus improve the current understanding of underlying biological processes[1–3].

Machine learning provides automation for analyzing big data, such as that acquired in large-scale, image-based experiments, and it has been successfully utilized for phenotypic analysis tasks[4]. Although a great variety of software tools are available for performing imaging assays in a supervised manner (e.g. Cell-Profiler Analyst, Ilastik, CellCognition, Advanced Cell Classifier[5]), all of them rely on the assumption that the underlying biological processes have stable steady states that can be dissected into discrete phenotypic classes (Fig. 1a). However, biological processes inherently contain continuous transitions between these phenotypes, consequently restricting the modelling to a set of discrete states reduces the potential to fully understand biological phenomena.

The application of traditional classification models for single-cell image analysis[6–8] is especially unreliable when the cells of interest change their morphological features gradually in the course of time (e.g. cell cycle). Annotation of such data is error-prone and laborious, and even field experts tend to make faulty

decisions (e.g. in the case of samples with interclass properties), often leading to arbitrary labelling. Additionally, user-defined classes may obscure the real underlying distribution by inappropriate discretization.

Currently, none of the available and widely used software tools enable single-cell-based image analysis in a continuous, supervised manner. Instead, unsupervised models, such as Lineage Reconstruction Techniques (LRT)[9] and Dynamic Time Warping (DTW) prevail. Cycler[8] is an LRT and embeds 5 pre-selected image-based single-cell features to a one-dimensional (1D) continuous space called the cell-cycle trajectory. Similarly, Cai et al. used DTW to align mitotic cells into the mitotic standard time based on 6 selected features[10]. HipDynamics is a software for visualizing cell population dynamics in live-cell imaging data and it utilizes unsupervised linear regression to characterize the changes in user-selected features[11]. Indeed, these tools provide robust solutions for their targeted tasks, but the lack of expert interaction significantly reduces the potential to customize these methods for various purposes. Therefore, another set of tools known as Visual Analytics (VA) was developed, offering various techniques for experts to interactively change the machine learning model through a visualization interface, which is most often a continuous space (visualization map)[12,13]. CellCognition was a pioneer of supervised tools, designed with the intent to efficiently analyze biological processes, however still using classification[7].

Here, we propose a methodology called Regression Plane (RP), an interface for fully supervised, continuous machine learning

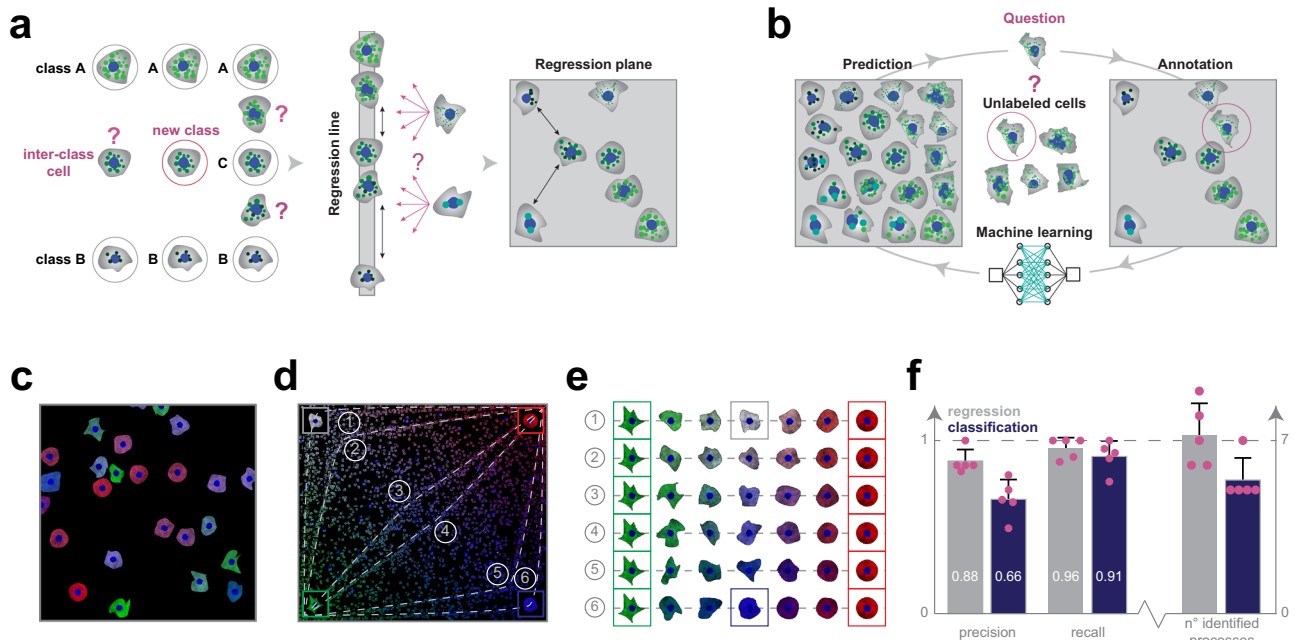

**Fig. 1 Classification vs regression. a** Regression plane concept. The classical way to model a biological process includes the phenotypical analysis of cells (i.e. subdividing cells into classes). However, in a high-content screening scenario, the multitude of different phenotypes makes it extremely challenging to create a set of representative classes. A possible solution builds on using a regression line, allowing to represent a single effect without the need of discretization. Nonetheless, biological processes are typically characterized by numerous ongoing effects. Thus, the regression plane represents a good trade-off between visualization capabilities and annotation complexity. Basically, it allows to represent a biological process with the limits of a planar graph. **b** Active regression. The aim of an active regression algorithm is to improve the training set (TS) to achieve better prediction performance. It is an iterative process where a cell that is difficult to annotate is proposed to the oracle who annotates it, and by doing so moves it to the TS used to train the regression model. **c** Synthetic dataset. Image from the synthetic dataset, generated using SIMCEP. **d** Experimental design. The designed processes overlayed on the space of perturbations. 6 processes are tracks in the space, and an extra process is formed of uniformly distributed cells (latent process 7). **e** Designed processes. The 6 continuous processes are modelled between two fixed endpoints: green cells of highly irregular shape and red, rounded cells. To assign a colour to the middle point of each process we interpolated between white (process 1) and blue (process 6). **f** Classification vs regression applied on synthetic data. Comparison of the performance of regression and classification. Statistics: precision, recall and the number of identified processes. Columns represent mean, error bars show the standard deviations from $n = 5$ independent users/experimental setup. Source data are provided as a Source Data file.

appropriate for image-based single-cell analysis. The idea originates from a study of an influenza A virus entry in which histone deacetylase-mediated reorganization of the microtubules led to various endosomal morphological and trafficking phenotypes that affected influenza infection[14]. The scatteredness of late endosomes and lysosomes (single output variable) was determined using regression instead of classification. Restricting the output to a single dimension prohibited the modelling of branching, circulating, parallel and crossing processes, therefore we extended the approach to utilize a 2D plane (Fig. 1a). Considering cellular steady states as graph nodes and gradual changes between the states as edges, the biological systems that correspond to planar graphs can be modelled with RP. Further extension of the modelling to 3D would increase the complexity of labelling and raise the chance of annotation errors. Additionally, to improve the quality of the annotated sets and decrease the time required from experts, we have incorporated active learning methods appropriate for regression-based phenotyping.

## Results

**Regression plane**. Regression plane is implemented as an open-source module of Advanced Cell Classifier (ACC)[6], and it has been available since ACC v3.0. RP was incorporated into traditional phenotypic classification in a hierarchical manner: each class may be extended with a distinct regression plane, allowing multiple regression planes to be included in a single project. RP is easy to use, well documented and supported by video tutorials (Supplementary Software 1, Supplementary Movies 1, 2). Annotation is performed by assigning continuous labels to representative cells via placing them on a 2D plane. After training, RP predicts the position of every unlabelled cell and outputs versatile and easy-to-read visual representations at single-cell, population and treatment levels (for details see the Methods section).

Similarly to classification, a representative Training Set (TS) is also essential for RP. Active learning algorithms are routinely used in classification to find the most efficient TS[15] but are not widely used in regression[16]. In this work, we introduce various active regression algorithms by extending those used in classical active learning tasks (Fig. 1b, Supplementary Fig. 1a). These methods propose cells whose automatic prediction on the regression plane is uncertain or ambiguous. Details are reported in the Methods section.

**Synthetic experiment: classification vs regression**. To analyze data discovery capabilities of RP, we generated a synthetic HCS image dataset simulating perturbations of cell shape and protein expression (Fig. 1c, Supplementary Software 2). We designed gradual perturbations to enable smooth transition between cell states and hence facilitating the modelling of biological processes. We defined 6 processes as continuous changes from one cell state to another, plus an extra process (latent process 7) formed from uniformly distributed cells (Fig. 1d, e). Each well in the HCS plate was associated with an underlying process, and the corresponding images were generated by sampling cells uniformly from the process distribution. Based on the associated processes we defined a partitioning on the wells (those wells were in the same partition that had the same process associated to them), forming the ground-truth of our experiment. Details about the modelled biological processes are reported in the Methods section.

Subsequently, ten microscopy experts were divided into two groups and asked to identify the distinct underlying processes in the experiment (Supplementary Note 1), or equivalently to define a partitioning on the wells. The first group of five experts used ACC v2.1 (extended with Supplementary Software 3 to compensate for the advanced clustering features available in RP)

to annotate cells with discrete labels, while the other group used RP only (ACC v3.0). Despite the great variety of the regression planes created by the microscopists (Supplementary Fig. 2), the results obtained using RP significantly outperformed the classification, both in terms of precision and recall (Fig. 1f). Specifically, the experts using RP performed better in estimating the number of ongoing processes, and achieved, on average, an improvement of approximately 20% in precision and 5% in recall, upon defining image sets containing cells with similar behaviour.

**Lipid droplet study**. Lipid droplets are storage units for neutral lipids, including triglycerides, and play a significant role in several disorders, including e.g. cardiovascular diseases. We evaluated whether siRNA perturbations of candidate genes, previously revealed to influence blood triglyceride (TG) levels in humans in a genome-wide association study[17], would affect the morphology of lipid droplets (LDs) in cultured hepatocytes (Huh7 cell line). Regarding their continuous changes in localization, number and size, LDs form a heterogeneous population reflecting different cellular metabolic states[18]. Thus, RP was used for the analysis of lipid droplets labelled with LipidToxGreen (Supplementary Fig. 3a–c), a probe used for quantitative analysis of neutral lipids. To train the model, 457 cells were placed on the regression plane by a cell biology expert (Fig. 2a).

We found that siRNA-mediated knockdown of TM6SF2 (a gene associated with decreased blood TGs) led to increased intracellular staining of neutral lipids, as it had been expected from the earlier evidence of TM6SF2 affecting hepatic lipid droplet content and TG secretion[19]. In contrast, the cells transfected with siRNAs targeting CD300LG (a gene associated with increased blood TGs[17]) showed a decreased amount of intracellular TGs, accompanied by the disappearance of (larger) LDs. Additional biochemical analysis measuring cellular TG levels confirmed these findings (Supplementary Fig. 3d). These data provide functional evidence for the role of CD300LG in regulating TG metabolism in hepatocytes.

Intriguingly, the knockdown of TM4SF5 (a gene associated with decreased blood TGs[17]) which codes for a protein functioning as an arginine sensor and mTORC1 regulator on lysosomal membranes[20], not shown earlier to affect TG levels in functional studies, promoted the increase of small LDs (Fig. 2b). Meta-visualization and clustering of the regression planes (Fig. 2c, Supplementary Fig. 3e–h) further supplemented the findings from an earlier study[17], and suggest that CD300LG and TM4SF5 may have biological effects on hepatic TG levels and LD composition, to be further addressed in future studies. Details are reported in Methods section.

**Time-lapse microscopy: cell cycle analysis**. We tested the capabilities of RP on 2 different time-resolved datasets. First, RP has been demonstrated to be capable of reproducing an unsupervised mitotic time model developed in the MitoCheck project (www.mitocheck.org).

Cai et al.[10] analyzed cell mitosis by performing time-lapse experiments to establish a canonical model for the morphological changes appearing during the mitotic progression of human cells. They reorganized the feature space according to the mitotic standard time instead of the imaging time (see Fig. 1f in ref. [10]), and by applying an unbiased peak-detection method in the warped feature space they identified up to 20 mitotic stages. The model was then used to integrate dynamic concentration data of several fluorescently labelled mitotic proteins, and to create a generic dynamic protein atlas of human cell division. Their public data include 3D images and segmented masks of 31 z-stacks. We intended to analyze this dataset without using prior feature

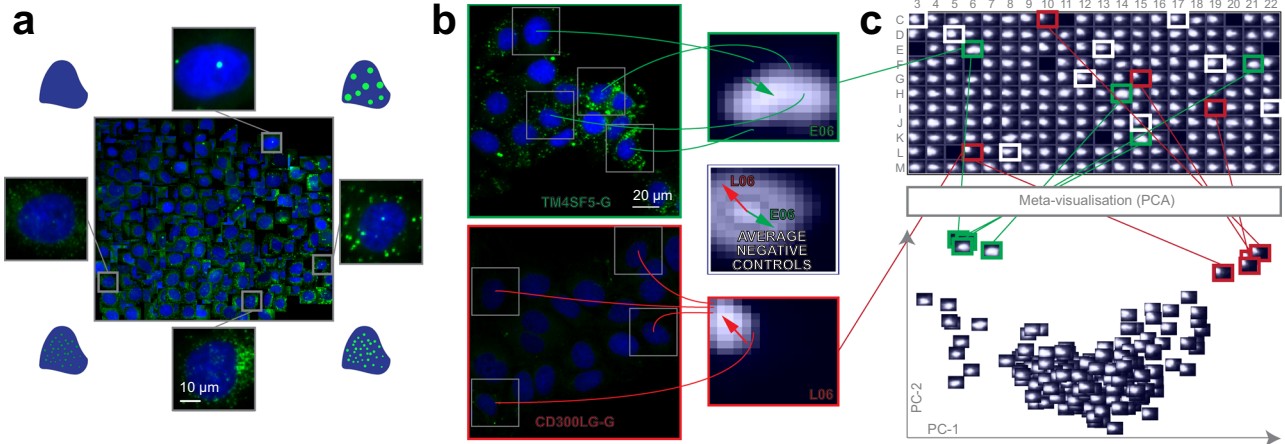

**Fig. 2 Lipid droplet dataset. a** Training set. Regression plane of 457 cells representing various lipid morphologies, created by an expert biologist. **b** RP output. Kernel Density Estimation (KDE)-maps of the predicted regression positions for cells treated with selected siRNAs. Arrows originate from the peak of the control KDE-map, and point to the peaks of the selected KDE-maps. **c** HCS analysis. Plate-based analysis performed by comparing well-based KDE-maps. Meta-visualization (in this case PCA–Principal Component Analysis) is obtained by extracting the principal components (PC1 and PC2) of the flattened KDE-maps.

information about the underlying process by exploiting regression techniques to characterize mitosis.

In our analysis, a field expert created a regression plane representing the process of mitosis, resulting in a training set of 585 cells (Fig. 3a). After prediction, the cells followed the designed circular path recalling canonical mitotic phases (Fig. 3b–d), while they also represented subtle phenotypic changes and single-cell differences in the regression plane. Additionally, we investigated whether the fluorescent tags have effect on the distribution of cells on the regression plane, and in most cases we did not observe undesired cellular behaviour due to the perturbations (Supplementary Fig. 4). Finally, we compared the results of the original methodology presented by Cai et al. (multi-dimensional dynamic time warping for creating the standard mitotic time, Fig. 3e) with the results obtained by RP (Fig. 3f), and we concluded that RP is capable of reproducing a mitotic time model equivalent to the original one. This indicates that RP can compete with complex analysis techniques, such as DTW. Moreover, RP provides the flexibility to customize the output space, enabling higher resolution analysis of user-defined sections of the biological process.

**Time-lapse microscopy: blood cell differentiation**. The fruit fly, *Drosophila melanogaster*, serves as a popular model system to study innate immune functions, such as phagocytosis, wound healing and capsule formation[21]. In the larva, these functions are executed by hemocytes, which are categorized into three main cell types: (1) phagocytic plasmatocytes, accounting for the majority of circulating hemocytes, (2) crystal cells, which play a role in melanization and wound healing, and (3) lamellocytes, which are large flat cells that appear only in certain tumorous genetic backgrounds or following immune induction[22]. Such an immune induction appears in nature as a result of egg-laying by a parasitoid wasp, *Leptopilina boulardi*. Following infestation, newly differentiating lamellocytes, together with plasmatocytes, eliminate the invader by forming a multilayer capsule around the wasp egg[23–25]. Lamellocytes are also produced when larvae are wounded with an insect pin[26] (Supplementary Fig. 5).

Cell lineage-tracing studies revealed that plasmatocytes, which had previously been considered as terminally differentiated phagocytic cells, show plasticity, and are capable of differentiating into encapsulating lamellocytes upon immune induction[22,27–29].

This transdifferentiation process has been underlined by recent single-cell RNA sequencing studies[30,31]. However, the cellular intermediates of the plasmatocyte-lamellocyte transition process have not been characterized morphologically in detail so far, and the routes of differentiation are still controversial[32]. A study by Anderl et al.[33] described two types of lamellocytes, and suggested that only the smaller type II lamellocytes (Supplementary Movie 3) differentiate from plasmatocytes, while the regular, flattened type I lamellocytes (Supplementary Movie 4) originate from dedicated precursors.

To clarify the potential routes of differentiation, we developed an ex vivo method for culturing Drosophila hemocytes, appropriate for monitoring their differentiation with time-lapse microscopy. Blood cell types can be characterized by their morphologies and in vivo transgenic reporter expression pattern[33]. The regression plane was manually trained using 109 cells based on their morphology and reporter gene expression (Fig. 4a).

The analysis revealed that 5.6% of the plasmatocytes transdifferentiated into lamellocytes upon immune induction (wounding) of the larvae (the threshold line is indicated in Fig. 4c), which is well reflected by the expression of cell type specific transgenes (Supplementary Movies 5, 6). After the formation of lamellocytes, no significant alterations in their cell size were observed, indicating that all types of lamellocytes are terminally differentiated cells. Most of the plasmatocytes (94.4%), however, did not differentiate into lamellocytes, but either spread out, increasing their cell size, or kept their size and morphology during the experiment, which is in line with the results of in vivo studies on blood cell differentiation in Drosophila.

However, in the case of lamellocytes instead of identifying 2 clearly separated subtypes I and II, we have observed that the differentiation processes are evenly distributed on the regression plane, as reflected by specific features (Fig. 4b, c, f). This finding suggests that type I and type II lamellocytes, both differentiating from plasmatocytes, are not definitely distinguishable cell types, but rather they are two extreme stages of a size continuum (Fig. 4e). Details are reported in the Methods section.

## Discussion

Regression plane increases the resolution of classification to represent subtle phenotypic differences by exploiting regression techniques, extended by active learning. First, using artificial

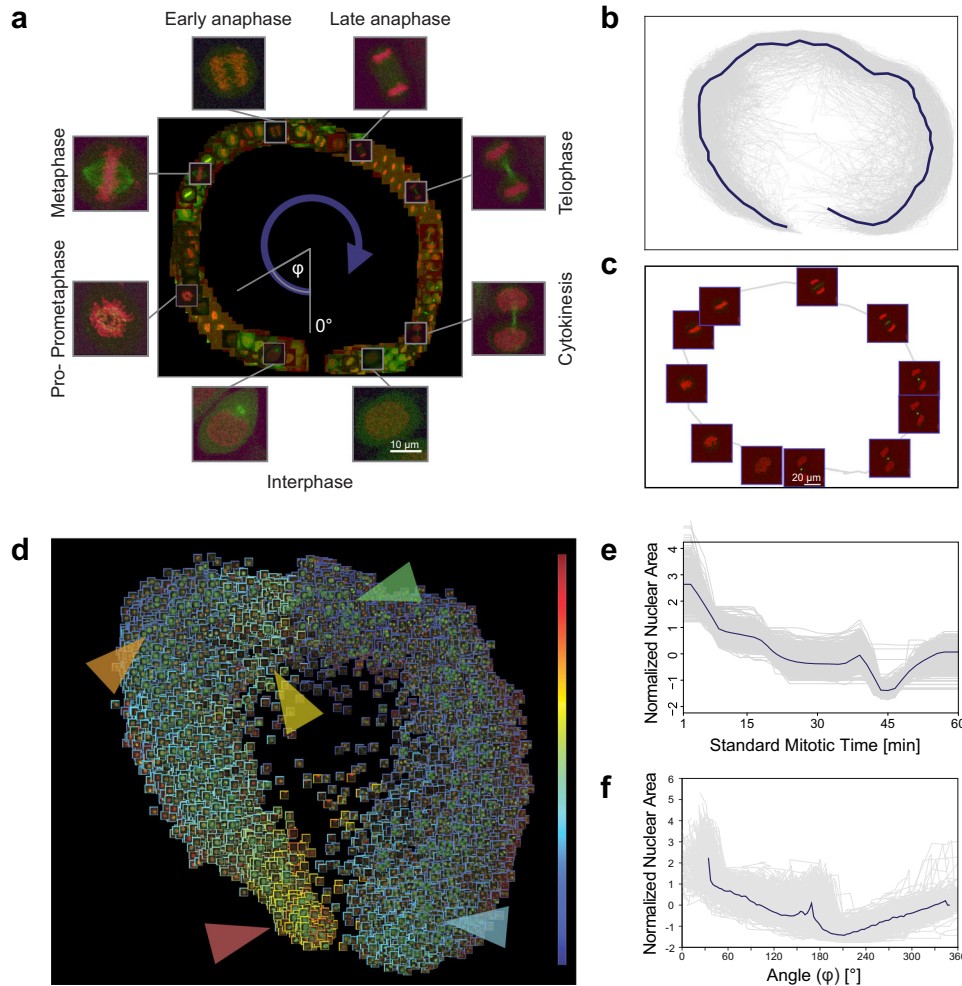

**Fig. 3 Mitosis data analysis. a** Regression plane of 585 cells annotated by a microscopy expert. **b** 498 trajectories for all the predicted cells. The median curve is shown in solid blue. **c** Example of a single-cell trajectory with representative cell icons visualized. **d** Regression plane with all ($n = 19,920$) predicted cells. The borders of the cell icons correspond to their nuclear area (Colour Frame module). Highlighted regions: early prophase region, large nuclear area (red). Metaphase region, nuclear area decreased (orange). Early-anaphase region, nuclear area is increasing as spindle fibres are pulling chromosomes apart (yellow). Anaphase, nuclear area dropped as the nucleus is considered as two separate objects with half the area (green). Late-telophase, nuclear area increasing up to half of the initial value (blue). **e** Trend for the normalized nuclear area according to standard mitotic time. Grey lines represent single-cell trajectories. **f** Trend for the normalized nuclear area according to the regression plane. Grey lines represent single-cell trajectories. The coordinates predicted by RP were converted to 1D by taking the angle argument of the polar coordinate representation as illustrated in **a**.

datasets we have demonstrated its capability to outperform the available classification tools in phenotypic discovery. Second, we have applied RP to analyze lipid droplets in hepatocytes during siRNA-mediated gene silencing, serving as a model of a heterogeneous population that reflects different cellular metabolic states. We have revealed genes playing a crucial role in regulating triglyceride levels in hepatocytes. Finally, we have identified the previously undiscovered continuous characteristics of hemocyte differentiation in *Drosophila melanogaster*. Our findings indicate that RP is a promising tool to explore biological data in a continuous manner, reflecting the non-discrete nature of biological processes.

## Methods

**Synthetic dataset**. To generate the dataset we used a customized version of SIMCEP[34], provided as Supplementary Software 2. Synthetic microscopy images were organized into a 24-well plate format, and the dataset was composed of 9 images/well, for a total of 216 images and 8117 cells. The images of each well were generated by considering a predominant process mixed with other ones. To model the continuous processes we fixed two endpoints: green cells of highly irregular shape, and red, rounded cells (Fig. 1e). The degree of cell shape deformation decreases from the green to the red endpoint. Next, for each process we selected a

middle point, and assigned a colour to that, ranging from white (process 1) to blue (process 6). The colour of the cells in each process was then defined by linear interpolation between the colour of the middle point and one of the two endpoints. The generated dataset was deposited to the Broad Bioimage Benchmark Collection (BBBC), and it is freely available at: https://data.broadinstitute.org/bbbc/image_sets.html (dataset ID: BBBC031).

**Lipid droplet dataset**. Huh7 hepatocellular carcinoma cell line (from Prof. Ilkka Julkunen, THL, Finland[35]) was authenticated using Promega StemElite™ ID System at Genomics Unit of Technology Centre, Institute for Molecular Medicine Finland (FIMM), University of Helsinki. The cells were cultured in Minimum Essential Medium (MEM, Gibco® Life Technologies) supplemented with 10% FBS (fetal bovine serum, Gibco® Life Technologies), 100 IU/ml Penicillin and 100 µg/ml Streptomycin (Penicillin/Streptomycin combination, Gibco® Life Technologies) at 37 °C incubator with 5% $CO_2$. siRNAs (Supplementary Data 1) were transferred from source plates (Echo Qualified 384-Well Low Dead Volume Microplate, 384LDV, Labcyte) to the assay plates (384 -Well Flat Clear Bottom Black Polystyrene TC-Treated Microplates, Corning®, USA) in a final concentration of 10 nM with Echo® 550 Liquid Handler (Labcyte, UK) and Echo Cherry Pick software (version 1.4.4). 25 nl/well of transfection reagent Lipofectamine RNAiMAX (Invitrogen, Life Technologies, USA) in 5 µl of Opti-MEM (Gibco® Life Technologies) was added to the assay plate with Multidrop Combi nL Reagent Dispenser (Thermo Fisher Scientific Oy, Finland). The cells (750 cells in 20 µl of complete medium/well) were delivered to the wells with Multidrop Combi Reagent Dispenser (Thermo Fisher Scientific Oy, Finland) using a standard cassette

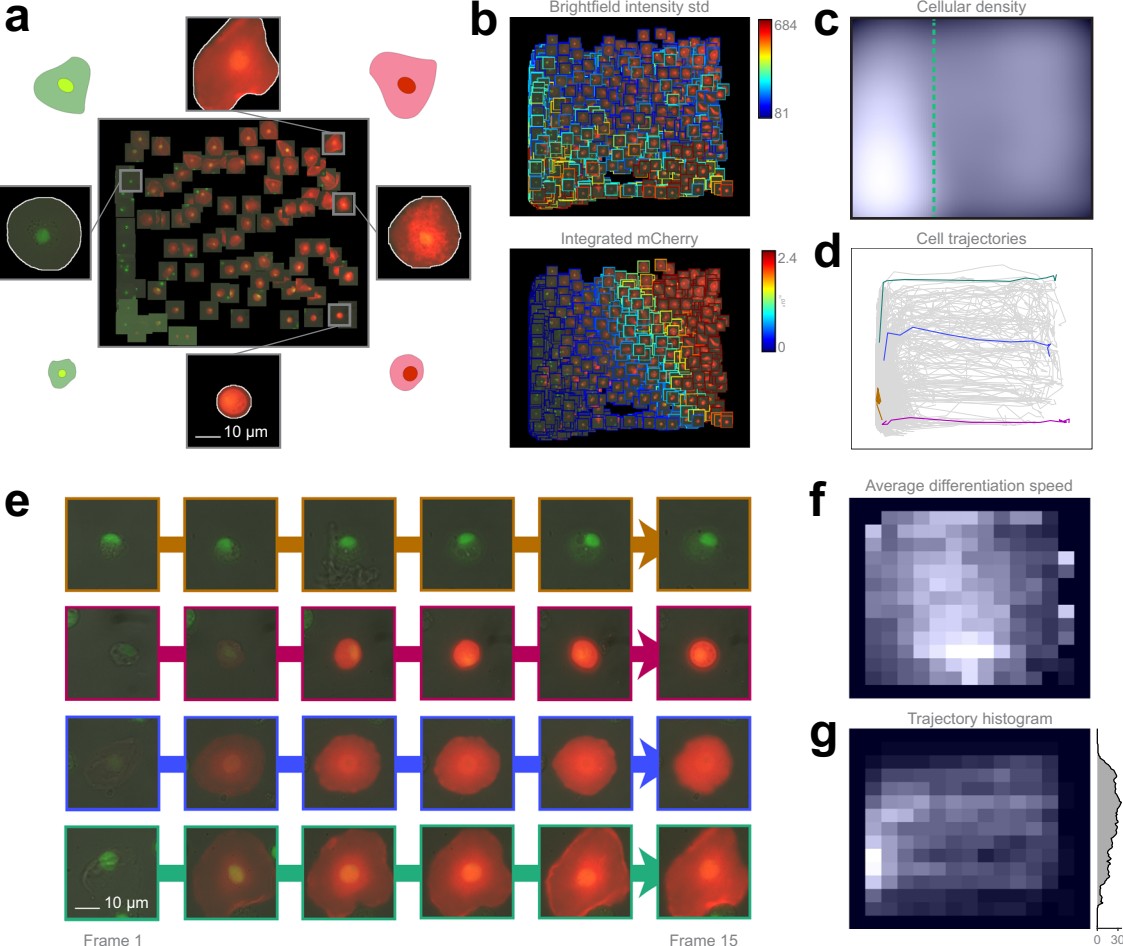

**Fig. 4 Hemocyte dataset analysis. a** Training set. 109 cells were placed on the regression plane by a microscopy expert. Cells were segmented by applying the NucleAIzer[40] deep learning method on brightfield microscopy images. **b** Single cell features. Colour-coded feature values overlay on the predicted cells. **c** Density plots. Kernel density estimation of single cells. **d** Single-cell trajectories. 2323 cell trajectories on the regression plane. **e** Selected cell trajectories. Representative phenotypes highlighted in **d**. **f** Differentiation speed histogram. Cell differentiation speed on the regression plane. **g** Trajectory histogram. 2D trajectory histogram on the regression plane and 1D projection with trajectory counts, including only those trajectories that reach beyond the green line in **c**.

(Thermo Fisher Scientific Oy). After 72 h of siRNA transfection the cells were fixed with 4% paraformaldehyde, quenched with 50 mM $NH_4Cl$ and stained with Lipidtox Green (HCS LipidTox Green Neutral Lipid Stain, Invitrogen) and 300 nM DAPI (Sigma-Aldrich) for 30 min at RT. Nine images/well were acquired per channel for duplicate plates with an automated epifluorescence ScanR microscope (Olympus) equipped with a 150 W Mercury-Xenon mixed gas arc burner, a 20× long working distance objective (UIS2) and a digital monochrome CCD camera (Hamamatsu). The image resolution was 1344 × 1024 pixel and 16 bit per channel. The 2 identical plates contained a total of 3956 images of 232,084 cells (>2200 cells per siRNA). The generated dataset was deposited to FigShare[36].

To validate our findings, additional biochemical analysis was performed to siRNA-transfected Huh-7 cells. The cells were collected in 0.2 N NaOH, followed by lipid extraction. TGs and CEs were resolved on TLC plates using hexane/diethyl ether/acetic acid (80:20:1) as the mobile phase. Lipids were visualized by charring, the plates were scanned and the intensities were quantified by ImageJ.

**Blood cell differentiation dataset**. Early third instar Me larvae (eaterGFP as a marker of plasmatocytes, MSNF9MOmCherry as a marker of lamellocytes[33]) were immune induced by wounding the cuticle with an Austerlitz Insect Pin® of 0.2 mm in diameter. Wounded larvae were kept on standard Drosophila food at 25°C. Circulating blood cells were isolated 12 h after wounding. Blood samples of 10 larvae were collected, pooled in 300 μl Schneider's medium (Lonza, Cat: 04-351 Q) supplemented with 10% fetal bovine serum (FBS; Gibco®, Cat: 10270) plus 0.01 mg/ml gentamicin (Sigma, Cat: G3632), 0.065 mg/ml penicillin (Sigma, Cat: P7794) and 0.1 mg/ml streptomycin (Sigma, Cat: S6501). Next it was spread into a well chamber of an 8-well μ-slide (Ibidi, Cat: 80826). Both sample storage and microscopic analysis were carried out at 25 °C.

We acquired 15-frame image sequences/field (141 fields) on 3 channels: brightfield, mCherry, and EGFP, with 2-hour-gaps between the subsequent frames. Images were acquired with a high-content screening microscope (Operetta, Perkin Elmer) equipped with a 60× high-numeric-aperture objective and a digital high resolution 14-bit CCD camera, yielding a total of 4230 images (2 plates, 2115 images in each). The image size was 1360 × 1024 pixels and 8-bit per channel, in TIFF format. The generated dataset was deposited to FigShare[37].

**Image segmentation and feature extraction**. In order to classify the cells in an image, ACC requires the position and features of each cell to be analyzed. For this purpose, we first flattened illumination distortions of the acquired images by using CIDRE[38]. Then, we used CellProfiler[39] and the NucleAIzer deep learning framework[40] to segment the cells and extract the standard features describing morphology, intensity and texture characteristics. Details of the image analysis and the regression models used in each experiment are reported in Supplementary Note 2.

**Regression models**. Regression methods, a subgroup of supervised machine learning techniques, are aiming at approximating continuous target variables. Alike for classification, various models have been proposed for regression, ranging from linear regression to neural networks and random forests[41].

The diverse set of regression models raise the problem of model selection for RP. As the RP is completely user-defined, it is impossible to have any prior assumptions on the function to be learnt, hence model selection should be data-driven. RP provides cross-validation assessment of model performance by root mean squared error measure (RMSE) and relative RMSE[42]. Additionally, two important aspects are to be considered when selecting the model.

First, the two-dimensional output format of RP requires the use of multi-target regression, as we require a 2D position (expressed by 2 coordinates) to be predicted. Traditionally, regression models aim at predicting a single continuous variable, which may be naturally extended for multiple dimensions by considering the outputs as independent variables, also called the single-target (ST) method[43]. On the contrary, it has been reported several times that multi-target models that exploit the possible correlation between the output variables may yield significantly better results than the ST methods[44,45]. Consequently, when a strong relationship between the output variables is evident, choosing a multi-target regression model is more appropriate.

Secondly, models that are capable of providing a probabilistic output (i.e. those that provide not only the predictive mean, but also some sort of uncertainty) are less wide-spread for regression than for classification. However, uncertainties provide valuable information to assess the model's performance, and most of the active learning strategies essentially rely on them.

Gaussian processes (GPs) can be used as non-parametric regression models with a probabilistic output[46]. Instead of providing a single prediction for each cell, GP returns a normal distribution whose mean can be used as the predicted value. More importantly, its variance is an estimate for the uncertainty of the given cell. GP is originally considered as a single-target method, however, its multi-target extensions also exist and are known as co-kriging[44,47]. Although GP is a non-parametric method (hence training is not required in principle), it still has hyperparameters (mean, covariance, likelihood, inference functions and their parameters) that can be optimized for enhanced performance. The most frequently applied iterative optimization methods (gradient descents) require initial hyperparameter settings which significantly affect the quality of the ultimate hyperparameter set. Consequently, we have designed heuristic hyperparameter initialization methods for several mean and covariance functions as described in Supplementary Note 3. Due to the broad selection of implementable models, RP provides an interface (via Object Oriented Programming) to facilitate the extension of implemented regression methods. By default, the package contains bridges to several models from Weka[48], Mulan[49] and Matlab's Deep Learning Toolbox. The full list and instructions on how to include new models are provided in Supplementary Note 4.

**Active regression.** Usually, the most time-consuming part of statistical learning for biomedical applications (including shallow and deep learning) is the procedure of annotation, and – as transfer learning is rarely used – it is often repeated for new experiments. Active learning[50] aims at reducing the number of training samples needed to achieve the most representative training set by automatically proposing cells for annotation. It has previously been shown by Smith and Horvath[51] that active learning reduces the time cost of annotation in HCS compared to classical labelling. Most of the active classification methods are based solely on the predicted class labels, enabling the underlying model to be freely modified. However, these methods are not directly applicable for regression, as they assume that the predicted label is discrete. Active regression methods were developed by Cohn et al.[52], based on variance reduction for Neural Networks, Mixture of Gaussians and Locally Weighted Regression. Here we present active regression methods inspired by the general active classification approaches, and a specific method for Gaussian Processes utilizing its properties (Supplementary Fig. 1).

*Committee members.* The Committee Members approach is inspired by the *QueryByCommittee* active classification method. Similarly to cross-validation, a set of models (committee) is built up from the available training samples, and a measure of disagreement is defined for the committee. In case of regression, the classical measures cannot be applied directly for two reasons: (1) they rely on the fact that the output is discrete, and (2) they require a probabilistic model. Thus, we propose using the quadratic mean of the Euclidean distance between the committee consensus and the single committee predictions. Hence, the next cell to be labelled by the expert is defined by the following formula:

$$x^* = \underset{x}{argmax} \sqrt{\sum_{i=1}^{C} \frac{d(\hat{y}_i, \bar{y})^2}{C}}$$ (1)

where $C$ is the size of the committee, $\hat{y}_i$ is the predicted position for $x$ (a sample not taken from the TS) by the $i^{th}$ committee member, $\bar{y}$ is the mean of $\hat{y}$, and $d$ is the Euclidean distance.

*Empty regions.* The Empty regions method targets the cells which were predicted to the least dense region of the regression plane in terms of training samples. This heuristic is supposed to explore those cell types that are not presented in the TS.

*Out of bounds.* By design, the regression plane is represented by a unit-square, and has limits in each direction. However, this limitation was not incorporated into the regression models, consequently it is possible that cells are predicted outside of the regression plane's boundaries. Therefore, we propose a strategy that selects these cells for annotation, ranked by their distance from the edges of the regression plane.

*Uncertainty sampling.* When a probabilistic regression model (such as GP) is available, then, instead of plain predictions, a posterior distribution is defined for each cell, enabling the application of active learning methods aiming at decreasing the variance of this posterior. Our proposed method targets the cell with the highest posterior variance, where the final value for the selection is determined by taking either the mean, the sum, the product, the minimum or the maximum of the 2 separate variances, calculated for each output dimension of the regression plane.

*Overall uncertainty sampling.* GP has an intriguing property, namely that the posterior distribution is independent of the actual TS positions; it only depends on the input features and the hyperparameters of the GP. In consequence, given fixed hyperparameters, it is possible to exactly calculate how the posterior variance changes, assuming that a new cell is included in the TS even without knowing its position on the regression plane. Executing this calculation for all possible candidates, the resulting cell proposed for annotation is the one that decreases overall variance the most. This approach is formulated by:

$$x^* = \underset{x}{argmin} \sum_{i=1}^{N} f_\sigma^x(x_i)$$ (2)

where $N$ is the size of the full dataset (including the training dataset) and $f_\sigma^x(x_i)$ is the variance for $x_i$, supposing that the GP was trained on the available training set extended with $x$. The predictive variances for individual samples are calculated from the diagonal elements of the predictive variance matrix according to ref. [46] by the following formula:

$$K(X_*, X_*) - K(X_*, X)K(X, X)^{-1}K(X, X_*)$$ (3)

where K is the kernel (covariance) function, $X$ is the feature matrix of samples not yet predicted and $X$ is the feature matrix of the training set's elements.

We assessed the performance of the proposed active learning methods with 4 regression models: Random Forest, Gaussian Process, Neural Network and Support Vector Machine; on 2 of our datasets: Lipid droplets and MitoCheck containing 457 and 585 annotated cells, respectively. In each scenario the experiment started with randomly isolating 1/3 of the available samples to a test set, leaving the remaining 2/3 in a pool. Then, 10 cells were randomly selected from the pool for initializing the training set, followed by iteratively extending it with 290 cells according to the active query strategy. In each iteration a regression model was trained, and the relative root mean square error (RRMSE) was calculated on the test set.

The results from 50 independent runs are displayed in Supplementary Fig. 1b, c. In all but one (Gaussian Process in the MitoCheck dataset) scenario there was at least one active learning technique that outperformed random sampling, despite the high variance of error values among different regression models. The Random Forest and Gaussian Process models achieved smaller RRMSE values than the other two methods inhibiting the active strategies' ability to significantly improve the performance in these cases. Still, the CommitteeMembers strategy resulted in the lowest average area under the curve value in 5 out of the 8 cases. We also note that although mean prediction error is the most widespread measure of active learning, other aspects of the model performance (e.g. model coverage) might be equally interesting for the users.

**Regression plane output.** RP provides output in various formats to satisfy the diverse needs of field experts (Supplementary Movie 2). The simplest output can be obtained by predicting an image in the main window of ACC, by clicking on a cell to see its raw regression plane position. Alternatively, in the regression plane one can select an arbitrary number of images, so that all cells in those images are going to be visualized on the regression plane with their icon at their predicted position. Importantly, these predictions can easily be added to the TS as well.

For well-based analysis, a multi-component report can be generated for each plate. The first component of the report is a pdf file containing a heatmap (simple cell count in a discretized regression plane) and a kernel density estimation (KDE) visualizing the distribution of cells on the regression plane in the particular well (Fig. 2b, c). Besides, the difference and the most dense position shift between single wells, and the average of user-defined control wells are also included.

Secondly, RP provides standard visualization tools (PCA, t-SNE[53] and NeRV[54]) for assessing the relationships among the wells. Each of these methods can generate the figure of Plot of plots (PoP; Fig. 2c, Supplementary Fig. 4a–c). In PoP each well is represented by its KDE/heatmap, and the distance between these representations corresponds to the difference between the wells' regression plane distributions (i.e. similar wells are close in PoP, whilst differing ones are farther from each other). In case of plates with higher well-numbers (e.g. 96 or 384) this may result in an overwhelmingly dense diagram, so the PoPs can be re-loaded to RP where they can be examined interactively. Importantly, in the RP-PoP, wells of similar perturbations (replicates) can be highlighted with colours. In addition to these tools for visualization, a clustergram can also be generated, providing a way to compare the perturbations by performing hierarchical clustering (Supplementary Fig. 3e–h, Supplementary Fig. 4d). The matrix in the middle of the clustergram visualizes pairwise Kullback–Leibler divergence (KLD) between the cell-number weighted average of the replicate wells. Clustering is performed on the pairwise KLD matrix with correlation as distance, and average linkage.

Additionally, RP enables the analysis of underlying image features by the Colour Frame (CF) module. CF works by visualizing the feature distribution of cells from the regression plane, using an artificial colour scale. In particular, the user selects a specific feature and adjusts the visualization settings to define a colour for each cell icon's frame in the regression plane. (Figs. 3d and 4b). Notably, CF can be used either for fine tuning of the TS, or for assessing features of interest after prediction.

Finally, the Trajectory Plot (TP) facilitates the assessment of live-cell data composed of time-resolved image sequences of the same fields. Organizing the corresponding single-cells into trajectories using the predicted coordinates of the regression plane enables the visualization of the dynamics of underlying processes (Fig. 4c–f). TP is a multifunctional visualization tool that facilitates a better understanding of the continuous aspect of biological processes and offers several possibilities to investigate cell fates or to compare the development of particular cells as a function of time. Filtering functions help to find subgroups of phenotypes with different behaviours. Interestingly, the dynamics of the process can be perceived by animating the evolution of trajectories (Supplementary Movies 7, 8). We note that the observed distances and the derived speed of motion in trajectories are completely user-defined, hence they should be interpreted relative to the designed training set. This is a general property of supervised methods and represents a trade-off between customizability and fully unbiased approaches.

**Reporting summary**. Further information on research design is available in the Nature Research Reporting Summary linked to this article.

## Data availability
Synthetic dataset: https://data.broadinstitute.org/bbbc/image_sets.html (dataset ID: BBBC031). Lipid droplet dataset: https://doi.org/10.6084/m9.figshare.c.5067638.v1[36]. Mitocheck dataset: http://www.mitocheck.org/mitotic_cell_atlas/downloads/v1.0.1/mitotic_cell_atlas_v1.0.1_fulldata.zip. The training set for the Mitocheck data generated in this study is available as Supplementary Data 2. Drosophila dataset: https://doi.org/10.6084/m9.figshare.c.5075093.v1[37]. Source data are provided with this paper.

## Code availability
RP is a new module of ACC (current version 3.1). ACC is written in MATLAB (The MathWorks, Inc., USA). ACC supports the most common image formats (e.g. tif, bmp, png) and it works under Windows 64-bit, Linux, and OS X environments. Source code and standalone versions (which do not require a MATLAB license), video tutorials, and help documentation files are publicly available at: www.cellclassifier.org. All the ACC materials are copyright protected and distributed under GNU General Public License version 3 (GPLv3). Further software involved in this study: CellProfiler v1 is available freely at: https://cellprofiler.org/previous-releases. The CIDRE framework is freely available at: https://github.com/smithk/cidre. The nucleAIzer pipeline source code is available at: https://github.com/spreka/biomagdsb. The experiments involving Matlab were conducted with Matlab v9.5.0.1298439 (R2018b). The data analysis involving ImageJ was conducted with version 1.49b.

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

## Acknowledgements

The authors thank Antti Lehmussola and Pekka Ruusuvuori (Tampere University of Technology, Finland) for the information provided about the SIMCEP software; Samuli Ripatti and Ida Surakka (FIMM, University of Helsinki, Finland) for their valuable comments on our experiments related to the genes associated with dyslipidemia; Anna Uro (Faculty of Medicine, University of Helsinki) for providing expertise in the biochemical quantification of lipid levels; Mariliina Arjama (FIMM, University of Helsinki, Finland) for technical expertise in cell culture; the FIMM High Throughput Biomedicine Unit for providing access to high throughput robotics and siRNA library and the FIMM High Content Imaging and Analysis unit for HC-imaging (HiLIFE, University of Helsinki and Biocenter Finland); Olli Kallioniemi (FIMM, University of Helsinki, Finland) for support in HC-imaging capabilities and funding; Gabriella Tick and Máté Görbe (BRC, Szeged, Hungary) for their help with the software documentation; the Finnish Grid and Cloud Infrastructure (urn:nbn:fi:research-infras-2016072533) for computational resources; Dóra Bokor (BRC, Szeged, Hungary) for proofreading the manuscript. A.Sz., B.T., A.B., E.M., Cs.M. and P.H. acknowledge support from the Hungarian National Brain Research Program (MTA-SE-NAP B-BIOMAG), from the LENDULET-BIOMAG Grant (2018-342), from the European Regional Development Funds (GINOP-2.3.2-15-2016-00006, GINOP-2.3.2-15-2016-00026, GINOP-2.3.2-15-2016-00037), from the H2020 (ERAPERMED-COMPASS, DiscovAIR) and from the Chan Zuckerberg Initiative (Deep Visual Proteomics). A.Sz. and E.I. acknowledges support from University of Helsinki (Centre of Excellence matching funds) and Academy of Finland (project 324929). V.P., L.P. and P.H. acknowledge support from the Finnish TEKES FiDiPro Fellow Grant 40294/13 and FIMM High Content Imaging and Analysis Unit (FIMM-HCA; HiLIFE-HELMI) and Biocenter Finland, Finnish Cancer Society, Juselius Foundation, Academy of Finland Centre of Excellence in Translational Cancer Biology, Kymenlaakso and Finnish Cultural Foundation. V.P. acknowledges University of Helsinki post-doctoral research project grant. F.P. acknowledges support from the Union for International Cancer Control (UICC) for a UICC Yamagiwa-Yoshida (YY) Memorial International Cancer Study Grant (ref: UICC-YY/678329). V.H. and I.A. acknowledge the Hungarian National Research Fund (OTKA NKFI-2 NN118207). V.H. acknowledges support from the National Research, Development and Innovation Office (OTKA K-131484). L.P. and J.P. acknowledge support from the Academy of Finland, decision numbers 295694, 313748, 327352 and 310552.

## Author contributions

P.H. conceived and led the project. A.Sz. developed the Regression Plane tool. A.Sz. and A.B. developed the trajectory tool. T.B. debugged and released the software. A.Sz., F.P., T.B., I.G.V., E.M., Cs.M., L.P., S.T., V.P. and V.H. designed and performed the experiments. A.Sz., F.P., A.B. and V.P. tested the software tool. E.I., Y.Y., I.A., J.P. and P.H. supervised the project. F.P. prepared the documentation and website. A.Sz., F.P., V.P., V.H. and P.H. wrote the manuscript. A.Sz., F.P. and I.B. prepared the figures included in the paper. All authors read and approved the final manuscript.

## Competing interests

The authors declare no competing isnterests.
