## [Peer Review File · Nature Communications]

REVIEWER COMMENTS

Reviewer #1 (Remarks to the Author):

I have read with attention the manuscript titled Regression plane concept: analysing continuous cellular processes with machine learning by Abel Szkalicity et al.

the conclusions are supported, the study is convincing, several examples of applications are included and I believe the work will definitely influence thinking in the field and be of great interest.

One valid point that could be achievable to move from description of cells to description of perturbation is that It would be great if the effect of the interference in the mitochek dataset could be plotted in terms of variation on the regression plane of specific perturbation.

The major innovation presented in the paper is the possibility to train a classifier for image analysis in a continuous rather than discrete way and this is unambiguously of great value and interest for the community. Several attempts of more limited scope in similar directions have been in our understanding proposed in the past and some of these may be cited in this work where the authors consider it appropriate.

See Figure 2 where each object is described by each of the five classifiers in <https://www.nature.com/articles/sdata201718>
Use of regression in segmentation and not classification such as <https://pubmed.ncbi.nlm.nih.gov/27256155/>
See recent application of semi-supervised methods such as <https://www.embopress.org/doi/full/10.15252/msb.20199083>

Comments around specific points in the manuscript:

page 3, large scale imaging scenarios (reword)

page 4, cell cycle is an immediate example to describe a continuous process

page 6 and figure 2D, what the ground truth is here needs to be better defined

Supplementary figure 2, again the ground truth is unclear

Supplementary figure 4, the data from mitochek are very interesting, it is unclear to me how the 60°C confinement is imposed in the plane between the different stages of the cell cycle

Supplementary figure 4c, do the highlight actually refer to Supp figure 4b?

Supplementary videos: there is a typo throughout (the word 'simultaneously')

while the videos in mitochek and drosophila with all cells are instructive, there are many videos with single cells that do not add in my opinion to the conclusion.

Reviewer #2 (Remarks to the Author):

Dear Authors,

The concept presented here is fundamentally simple but novel idea. It is very smart and the great attractiveness of it is that it works, as the examples demonstrate very well. The paper is also very well prepared. Unfortunately I did not have time to test the provided implementation myself, which I would have liked to do.

I do find that the paper is a bit imbalanced in that so much of the interesting material is in the supplements. This is the result of length restrictions and I do not see an obvious solution to it, unfortunately. I can only hope that readers will access the supplements.

Some remarks, but they are of minor importance:

The second paragraph of the Main Text contains the statement that "biological processes are inherently continuous and modelling them as discrete states may reduce the potential to properly understand biological phenomena." Perhaps this is a bit too bald; it has been well argued that cells have discrete steady states, with of course developmental trajectories in between, which are important to understand. I feel that some of the supplementary videos illustrate that nicely. So maybe that could be rephrased.

Both the figures and the Supplementary Figure 4d prompt the reader to wonder whether distance on the RP and speed of motion on the RP are meaningful metrics. Is this something that can be usefully commented on, maybe in the supplement if there is no space in the paper? Does it depend entirely on how the training set is constructed, or not? Is this then a potential disadvantage of supervised versus unsupervised models, which can construct their own pseudo-time?

Minor quibble: The introductory slides to several videos spell "simultaneously" with an "i" instead of an "e". A few of these introductory slides should be shown a little longer, due to the amount of text.

Overall, an excellent paper and definitely a useful enabler for people involved in imaging applications. I am of the opinion that it meets all the criteria for acceptance.

Best Regards,

Emmanuel Gustin

REPLY TO REVIEWERS

We thank the Reviewers for the many valuable comments that helped us to improve our work. What follows is our detailed point-to-point reply to the comments received.

Hereafter, “C#” stands for Reviewer’s comments and “R#” for our replies. All the reference numbers (pages, citations, etc.) refer to the revised version of the text, unless explicitly defined otherwise. In the manuscript, all changes are reported with track changes.

COMMENTS OF REVIEWER1

[C1.1]

I have read with attention the manuscript titled Regression plane concept: analysing continuous cellular processes with machine learning by Abel Szkalicity et al.

the conclusions are supported, the study is convincing, several examples of applications are included and I believe the work will definitely influence thinking in the field and be of great interest.

One valid point that could be achievable to move from description of cells to description of perturbation is that It would be great if the effect of the interference in the mitochek dataset could be plotted in terms of variation on the regression plane of specific perturbation.

[R1.1]

We thank the Reviewer for his suggestion to extend our figures. The specific Mitochek dataset (Cai et al., Nature 2018) we analyzed involved only endogenous GFP tagging, consequently the perturbations were not expected to modify the distribution of the cells, otherwise the tagging itself would have had an undesired off-target effect by modifying the pace of the mitosis (i.e. cells may have spent more time in a specific mitotic stage due to the tagging). Investigating whether the fluorescent tagging has effects on the cell-density over the course of mitosis is straight-forward using the Regression Plane’s basic toolset. We now did this experiment. We considered each GFP tagging as a treatment applied to the cell population, and applied our unsupervised exploratory toolset (NeRV, PCA, t-SNE and hierarchical clustering) to reveal perturbations that significantly alter the distribution of the cell population. Our results (Supplementary Figure 4) indicate that some tagging methods (especially CEP192) show different cell distribution on the regression plane than other taggings, suggesting that they are interfering with mitosis to some extent. However, verifying these effects is out of the scope of our study.

[C1.2]

The major innovation presented in the paper is the possibility to train a classifier for image analysis in a continuous rather than discrete way and this is unambiguously of great value and interest for the community. Several attempts of more limited scope in similar directions have been in our understanding proposed in the past and some of these may be cited in this work where the authors consider it appropriate.

See Figure 2 where each object is described by each of the five classifiers in

<https://www.nature.com/articles/sdata201718>

Use of regression in segmentation and not classification such as <https://pubmed.ncbi.nlm.nih.gov/27256155/>

See recent application of semi-supervised methods such as

<https://www.embopress.org/doi/full/10.15252/msb.20199083>

[R1.2]

We thank the Reviewer for drawing our attention to these papers. After careful considerations, we cited the paper from Kerz *et al.*, as it is inherently related to continuous analysis via live-cell imaging. On the other hand, although

the works of Pascual-Vargas *et al.* and Sailem *et al.* build on image-based machine learning techniques, they both utilize (and ingeniously modify) the traditional classification setup. Hence we think that citing these works would be inappropriate.

[C1.3]

Comments around specific points in the manuscript:
page 3, large scale imaging scenarios (reword)

[R1.3]

We have changed the word scenarios to technologies.

[C1.4]

page 4, cell cycle is an immediate example to describe a continuous process

[R1.4]

We have included the cell cycle as an example when we introduce the concept of continuous processes.

[C1.5]

page 6 and figure 2D, what the ground truth is here needs to be better defined

[R1.5]

We thank the Reviewer for pointing out this ambiguity around the ground truth of the synthetic data. We revised our text describing the experiment and modified the figure caption to clarify our experimental setup.

[C1.6]

Supplementary figure 2, again the ground truth is unclear

[R1.6]

We have changed the figure caption and the legends in the figure to clarify this ambiguity, and to match the terminology with our revised text (response to [C1.5])

[C1.7]

Supplementary figure 4, the data from mitocheck are very interesting, it is unclear to me how the 60°C confinement is imposed in the plane between the different stages of the cell cycle

[R1.7]

We thank the Reviewer for drawing our attention to this possible confusion. The arc and the angle label was included in a) to facilitate the understanding of the polar coordinate angle representation in d). We have replaced the exact 60 degree confinement to a general phi representation to clarify the meaning of the arc.

[C1.8]

Supplementary figure 4c, do the highlight actually refer to Supp figure 4b?

[R1.8]

The highlights in Supp. Fig. 4c are independent from Supp Figure 4b. Supp. Fig. 4b shows a single-cell trajectory with cell icons to facilitate the interpretation of the trajectories. On the other hand Supp.Fig 4c shows general properties of the regression plane created to analyse mitosis. We have reorganized our figures in the revised manuscript.

[C1.9]

Supplementary videos: there is a typo throughout (the word 'simultaneously')

[R1.9]

We thank the reviewer for noticing our typo, we have corrected this mistake in all the videos.

[C1.10]

while the videos in mitocheck and drosophila with all cells are instructive, there are many videos with single cells that do not add in my opinion to the conclusion..

[R1.10]

We are delighted to hear that our conclusions were already well supported by our figures and videos with all cells. We combined the single cell videos to a montage.

COMMENTS OF REVIEWER2

[C2.1]

Dear Authors,

The concept presented here is fundamentally simple but novel idea. It is very smart and the great attractiveness of it is that it works, as the examples demonstrate very well. The paper is also very well prepared. Unfortunately I did not have time to test the provided implementation myself, which I would have liked to do.

I do find that the paper is a bit imbalanced in that so much of the interesting material is in the supplements. This is the result of length restrictions and I do not see an obvious solution to it, unfortunately. I can only hope that readers will access the supplements.

[R2.1]

We have reorganized our text by moving content from the Supplementary to the main text. In particular we moved the introductory sentences of the Synthetic, Lipid droplet and Blood cell differentiation datasets, and the whole paragraph the MitoCheck dataset into the Results section from the methods. We modified our figure layout and distribution to be better in line with the amount of text. We left the more technical descriptions in the Methods section.

[C2.2]

Some remarks, but they are of minor importance:

The second paragraph of the Main Text contains the statement that "biological processes are inherently continuous and modelling them as discrete states may reduce the potential to properly understand biological phenomena." Perhaps this is a bit too bald; it has been well argued that cells have discrete steady states, with of course developmental trajectories in between, which are important to understand. I feel that some of the supplementary videos illustrate that nicely. So maybe that could be rephrased.

[R2.2]

We thank the Reviewer for noting our indeed bit exaggerating wording in that sentence, we have rephrased it to be more moderate.

[C2.3]

Both the figures and the Supplementary Figure 4d prompt the reader to wonder whether distance on the RP and speed of motion on the RP are meaningful metrics. Is this something that can be usefully commented on, maybe in the supplement if there is no space in the paper? Does it depend entirely on how the training set is constructed, or not? Is this then an potential disadvantage of supervised versus unsupervised models, which can construct their own pseudo-time?

[R2.3]

We thank the Reviewer for his valuable questions and thoughts. The outcome of the regression plane analysis is entirely dependent on the training set, more precisely on the relation of the training set and the data we predict. However, this is a fundamental property of all supervised techniques, hence discussing its relevance is out of the scope of our article. In brief, all supervised models introduce the bias of the annotator in change for the increased customizability of the analyses. We have added a short discussion on these thoughts to the description of the trajectory plot in the methods section.

[C2.4]

Minor quibble: The introductory slides to several videos spell "simultaneously" with an "i" instead of an "e". A few of these introductory slides should be shown a little longer, due to the amount of text.

[R2.4]

We thank the reviewer for noticing our typo, we have corrected this mistake in all the videos. Additionally, we modified the timing of the introductory slides to give more time for the readers to familiarize themselves with the upcoming content.

[C2.5]

Overall, an excellent paper and definitely an useful enabler for people involved in imaging applications. I am of the opinion that it meets all the criteria for acceptance.

Best Regards,

Emmanuel Gustin

[R2.4]

Dear Dr. Gustin, we sincerely thank you for your comments and for your positive feedback on our work.

REVIEWERS' COMMENTS

Reviewer #1 (Remarks to the Author):

The manuscript is significantly improved and all my comments have been addressed.